# Dynamic light manipulation via silicon-organic slot metasurfaces

Tianzhe Zheng[1,6], Yiran Gu[2,6], Hyounghan Kwon ●[1,3,4], Gregory Roberts[1,5] & Andrei Faraon ●[1,3] ✉

Active metasurfaces provide the opportunity for fast spatio-temporal control of light. Among various tuning methods, organic electro-optic materials provide some unique advantages due to their fast speed and large nonlinearity, along with the possibility of using fabrication techniques based on infiltration. In this letter, we report a silicon-organic platform where organic electro-optic material is infiltrated into the narrow gaps of slot-mode metasurfaces with high quality factors. The mode confinement into the slot enables the placement of metallic electrodes in close proximity, thus enabling tunability at lower voltages. We demonstrate the maximum tuning sensitivity of 0.16nm/V, the maximum extinction ratio of 38% within ±17V voltage at telecommunication wavelength. The device has 3dB bandwidth of 3MHz. These results provide a path towards tunable silicon-organic hybrid metasurfaces at CMOS-level voltages.

Relying on sub-wavelength nanostructures, metasurfaces have been shown as promising candidates for replacing conventional free-space optical components by arbitrarily manipulating the amplitude, phase, and polarization of optical wavefronts in certain applications[1–3]. In recent years, the scope of their applications has been expanded towards complete spatio-temporal control through the introduction of active metasurfaces. These developments open up exciting new possibilities for dynamic holography[4], faster spatial light modulators[5], and fast optical beam steering for LiDAR[6]. Large efforts have been channeled into various modulation mechanisms[7]. Microelectromechanical and nanoelectromechanical systems (MEMS and NEMS)[8–11] have the advantages of low-cost and CMOS-compatibility, but the speed is limited up to MHz. Phase-change materials[12–14] have fast, drastic, and non-volatile refractive index change, but lack continuous refractive index tuning and have a limited number of cycles constraining applicability to reconfigurable devices. Through molecule reorientation, liquid crystal can have index modulation over 10%, while under relatively low applied voltages Tunable liquid crystal

metasurfaces, U.S. patent number 10,665,953 [Application Number 16/505,687][15]. Techniques of liquid crystal integration have also advanced after decades of development. However, the tuning speeds are limited to kHz range[16]. Thermal-optic effects can induce relatively large refractive index changes[17,18], but the speed is inherently limited and the on-chip thermal management can be challenging. The co-integration of transparent conductive oxide and metallic plasmonic structures[5,6] has been demonstrated in epsilon-near-zero (ENZ) regime to control the wavefront of reflected light, but the low reflection amplitude induced by the optical loss of the materials and the ENZ regime is unavoidable.

In modern photonics, a multitude of technologies for tunable optics and frequency conversion[19,20] are realized with nonlinear materials that have low loss and a strong $\chi^{(2)}$ effect, such as lithium niobate[21,22], aluminum nitride[23], and organic electro-optic (OEO) materials[24]. Their ultrafast responses make it possible to use RF or millimeter-wave control[25]. Developments in computational chemistry have also led to artificially engineered organic molecules that have

[1]T. J. Watson Laboratory of Applied Physics and Kavli Nanoscience Institute, California Institute of Technology, 1200 E. California Blvd., Pasadena, CA 91125, USA. [2]Department of Applied Physics and Material Science, California Institute of Technology, 1200 E. California Blvd., Pasadena, CA 91125, USA. [3]Department of Electrical Engineering, California Institute of Technology, 1200 E. California Blvd., Pasadena, CA 91125, USA. [4]Present address: Center for Quantum Information at Korea Institute of Science and Technology, 5, Hwarang-ro 14-gil, Seongbuk-gu, Seoul, Republic of Korea. [5]Present address: Tech4Health Institute, New York University Langone Health, New York, NY 10016, USA. [6]These authors contributed equally: Tianzhe Zheng and Yiran Gu. ✉e-mail: faraon@caltech.edu

record-high nonlinear coefficients with long-term and high-temperature stability[26,27]. However, their potential in modifying free-space light has been relatively unexplored until recently. Several OEO material-hybrid designs have demonstrated improved tunability of metasurfaces[28–30]. Utilizing dielectric resonant structures and RF-compatible coplanar waveguides, a free-space silicon-organic modulator has recently accomplished GHz modulation speed[31]. However, all demonstrations to date require high operating voltages ± 60V, due to low resonance tuning capability (frequency shift / voltage), which hinders their integration with electronic chips.

In this work, we propose combining high-Q metasurfaces based on slot-mode resonances with the unique nano-fabrication techniques enabled by OEO materials, which drastically reduces the operating voltage. The low voltage is mainly achieved from the ability to place the electrodes in close proximity to each other while hosting high-Q modes in between and the large overlap of the optical and RF fields in OEO materials. In the following sections, we first provide the design concepts and considerations for achieving a reduced operating voltage. Next, we numerically demonstrate the advantage of a particular selected mode compared to other supported modes in the structure. Finally, we experimentally realize our concepts and characterize the performance of the electro-optic metasurface.

## Results

The reported device and its operation scheme are depicted in Fig. 1. Light polarized along $x$ ($E_x$) is incident onto the device along $-z$ direction, and then couples into the slot mode hosted in between the silicon nano-bars. Gold electrodes are placed on top of the nano-bars and doped silicon is used to maximize the voltage drop across the slots filled with the OEO material. The active OEO material regions have nonlinear coefficients $r_{33}$ with each two adjacent slots exhibiting opposite signs of nonlinear coefficients due to the poling field

direction. When the operating signal is applied, the active layer induces a refractive index change

$$\Delta n(t) = -\frac{1}{2} n_{mat}^3 r_{33} E_{ext}(t) \qquad (1)$$

where $E_{ext}$ is the external electric field in the OEO material. Notice that due to the geometry of the electrodes, the signal electric fields $E_{ext}(t)$ also have opposite signs in adjacent gaps, as shown in Fig. 1. Therefore, the overall responses from any two adjacent slots are the same. Besides the operating voltage or the field in slots, the perturbation strength to the optical mode also depends on the overlap factor $\Gamma_c$ between the electric field profile and the optical mode profile. $\Gamma_c$ is irrelevant to the external voltages. To calculate this overlap factor, we need to treat $E_{ext}$, $r_{33}$ as spatial dependent variables. Based on cavity perturbation theory[32] (See the full derivation in supplementary section 2), we could extract the formula to calculate the shift of the resonant frequency:

$$\Delta\omega = -\frac{\Delta n_{avg}}{n_{mat}} \omega \Gamma_c \qquad (2)$$

where $\Delta n_{avg} = -\frac{1}{2} n_{mat}^3 r_{33} \frac{V_{ext}}{w_g}$ denotes the refractive index change upon applying a constant field of $\frac{V_{ext}}{w_g}$ across the gap.

By examining Equation (2), we can gain valuable insights into two distinct methods for reducing the voltage of the device: reducing the distance between electrodes and increasing the overlap factor. However, introducing closer metallic electrodes leads to inevitable losses, thus reducing the quality factors and limiting the sensitivity to refractive index changes. Therefore, transmitting electric fields through conductive dielectrics is preferred. In our reported device, doped silicon acts as the electrodes with gap width ($w_g$) down to 100nm. At the same time, this gap between the silicon nano-bars will

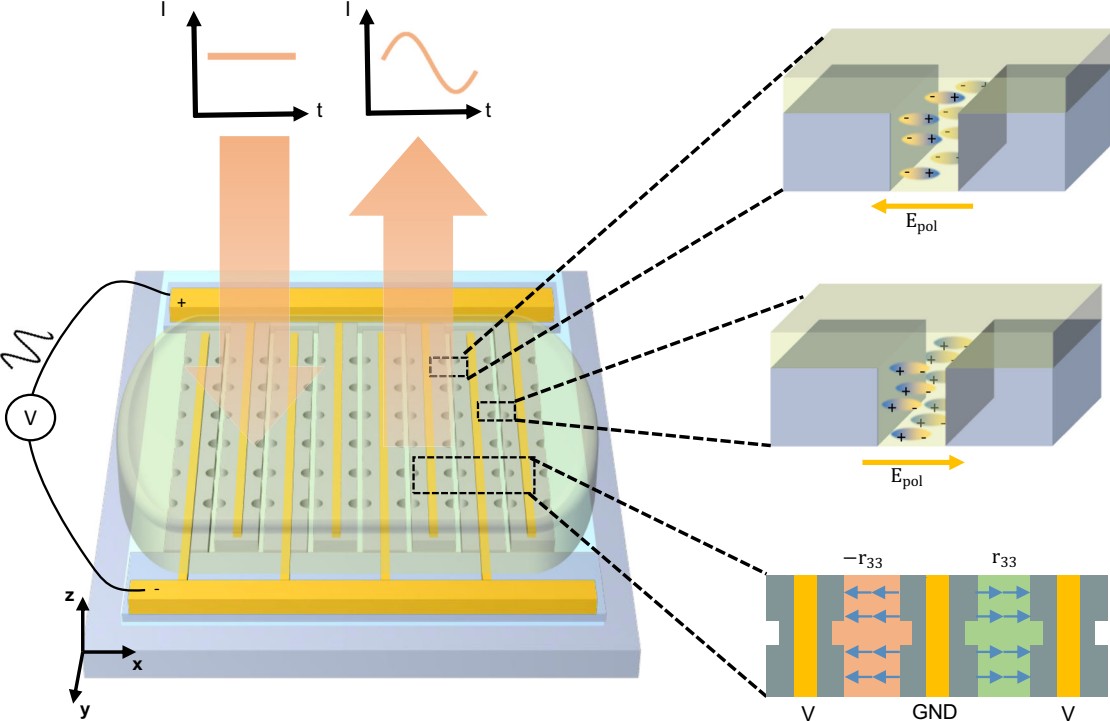

**Fig. 1 | Conceptual schematic of silicon-organic electro-optic tunable metasurfaces.** A beam of light is incident on the metasurface, which consists of silicon nano-bars. The light is coupled into the slot mode inside the metasurface, which is sensitive to any refractive index perturbation in the slots. The OEO material is coated on top of the metasurface and fills the slot waveguide between the silicon nano-bars. The organic molecules inside the slots are aligned with the DC/RF field generated by the electrodes. When the RF bias voltage is applied on the electrodes, the electro-optic (Pockels) effect will generate refractive index modulation. As a result, the intensity of the reflected beam will be modulated accordingly.

host slot modes[33,34] whose demonstrated high overlap with the OEO material has been utilized in many integrated silicon-organic modulators[35–37]. However, the slot waveguide is intrinsically decoupled from the free-space light due to momentum unmatching. To enable coupling with normally incident light, we create periodic notches along every slot[38,39]. The notch periodicity and the notch size dominantly determine the resonance wavelength and the coupling strength of the slot resonance, respectively. As a result, both the quality factor and the resonant wavelength could be judiciously engineered (see supplementary sections 1–2). Although a similar structure has been proposed for sensing[40], the design strategy and target applications here are completely different.

To show the advantage of the slot modes while shrinking the distance between electrodes down to 100nm, it is worth discussing the possible optical modes in such a structure. The detailed schematic view, top view, and cross-section view of the example device are shown in Fig. 2a, where three different colors (gray, blue, green) indicate different materials (silicon, silica, OEO material or HLD). In the numerical simulations shown in Fig. 2b–h, HLD has a refractive index of 1.85[27]. Figure 2b shows the poling field profile upon applying a DC bias across the electrodes. The x-z cross-section is cut at the center of a notch pair shown as dashed rectangle in the schematic view of Fig. 2a. Neglecting any interface effect[41], we can treat the relative amplitude of $r_{33}$ at each spatial point as following this pictured poling field profile. The geometry of the electrodes results in a high $E_x$ field along the slot and a rapid field decay above and below the slot. Therefore, to simplify the simulation, we assume that only the OEO material inside the slot is nonlinearly active. The structure will host various optical modes, many of which have a significant $E_x$ field component such that it could strongly overlap with the incident $E_x$ beam. Figure 2c–e show three cross-sectional optical mode profiles (Mode I, II, III) originating from different parts of the device. The cross-sections are cut at the same y position as in Fig. 2b. Mode I is the slot mode, which has the field highly confined inside the slot, as discussed above, even upon applying the notch perturbation. Mode II is the bound state guided within the slab. Notice that besides the slot mode, periodic notches also unlock the free-space radiation for this guided mode in the slab[42]. Mode III is a guided mode in OEO material, which has also been reported in[29]. Unlike the other optical modes, the field in slot mode is well aligned with the poling field and thus has the highest overlap factor $\Gamma_c$. As a result, under the same bias voltage, the slot mode has the largest resonance shift, as demonstrated through the simulation results in Fig. 2g–i. With the same amount of index change in the active region, Mode I, II, and III have resonance shifts of 2.73nm, 0.29nm, and 0.46nm, respectively. A more accurate model estimates the overlap factor by considering the orientation of the nonlinearity[29] (see supplementary material section 2). The calculated $\Gamma$ based on this model for modes I, II, and III are 0.156, 0.017, and 0.015, respectively. The slot mode shows an order of magnitude higher $\Gamma_c$, compared to the others. Therefore, the slot mode is crucial for low-voltage modulation in silicon-organic metasurfaces.

We experimentally realize the concept discussed above, using a silicon-on-insulator (SOI) wafer. The device's cross-section, top view, and voltage setting are schematically illustrated in Fig. 3a, b. The detailed device parameters are shown in supplementary material section 7. The nanostructures are fabricated with conventional nanofabrication techniques (see Methods section for details on the fabrication procedure). The step-by-step zoom-out scanning electron microscopy (SEM) images (prior to spin-coating of the OEO material) are shown in Fig. 3c–f. Doped silicon nano-bars have a resistance of 1 ∼ 10Ω · cm, and ∼100nm wide gold strips along the nano-bars are added to further reduce the voltage drop across the silicon. In Fig. 3c, the gold strips are deliberately aligned at the center of the silicon rail so that only minimal absorption is introduced (See supplementary section 3 and Fig. S3). Figure 3e shows an 80 × 100 μm² device. Multiple devices are fabricated on a chip as shown in Fig. 3f for increasing

the tolerance of the fabrication errors and testing multiple geometric parameters. After the coating of the OEO material, the device is wire bonded to a customized printed circuit board for poling and operating, shown in Fig. 3g.

To experimentally verify the relationship between resonant optical characteristics of the slot modes and the geometry of the device, we fabricated several devices having different design parameters and compared their measured optical properties with corresponding numerical calculations (see methods, supplementary section 6, and Fig. S7 for details in optical measurement setup). Figure 4 shows the calculated and measured spectra of the slot mode resonances with different geometries. We characterize the slot resonances by varying the notch period and notch size in Fig. 4a, b and c, d, respectively. In Fig. 4a, b, we observe that a 20 nm increment in notch periodicity leads to ∼21.6 nm and 22.2 nm average redshift of the resonance in simulation and experiment, respectively. Also, with respect to the resonant wavelengths, the resonance amplitudes, and the spectral shapes of the resonance, the measured spectra in Fig. 4c show good agreement with the calculated spectra in Fig. 4d. In particular, the quality factor increases with the decrease of the resonance amplitude. This trade-off is mainly due to the decreasing radiation rate to the top port (to +z direction), which results in the under-coupling between the slot mode and the illuminated light from the top[18]. Specifically, the amplitude of the resonance is determined by the ratio of the mode coupling rate between the input light and the slot mode to the sum of other undesired decay rates[43]. The undesired decays include absorption in the gold layer, scattering from the rough sidewalls or finite edges of the chip, and radiation to the oxide layer or the silicon substrate. As the absorption in the gold layer and the radiation to the silicon substrate are nearly inevitable in the proposed planar structures, the trade-off between the resonance amplitude and the quality factor is inevitable, especially when the coupling rate decreases. The proposed devices in Fig. 4e can achieve modulation amplitude over 10% and Q-factor over 1000 in the experiment even with the absorption in the gold.

Operation results under DC bias are shown in Fig. 5. Figure 5a–c show the results from 3 different devices under maximum bias voltages before any dielectric breakdown. The variation in the breakdown voltages results from the quality of the OEO material preparation and the device fabrication. The maximum absolute frequency shift of 5.5nm is achieved under ±17V in Fig. 5a, and the spectral shift per unit external DC bias is $S_{abs} = \Delta\lambda/\Delta V = 0.161$nm/V, which is ∼1.6 × higher than that of the previously reported tunable free-space optical modulators[31]. Using Eqs. (1) and (2), the in-device $r_{33}$ is calculated as 45.7 pm/V at 1495nm. In Fig. 5b and c, 2.3 nm and 2.6 nm resonance shifts are observed with ±11V and ±12V bias voltages, respectively. Figure 5b and c show high Q-factors over 1000. The increase in Q-factor also improves the normalized modulation figure-of-merit ($S_n = \Delta\lambda/(FWHM \cdot \Delta V)$[44]. In our best-performing device shown in Fig. 5b, $S_n$ is 0.09/V, which is an order of magnitude higher than other reported devices[44]. The reflection spectra are plotted in Fig. 5d as a function of different bias voltages. The spectra clearly show the bidirectional linear relationship between the bias voltage and the resonance shift, confirming that the spectral shift results from the electro-optic effect[29]. In Fig. 5e, the relative modulation ratio, $\Delta R/R$, from the device in Fig. 5c is plotted. The maximum modulation ratio is over 40%. It is worth noting that the asymmetry of the modulation is due to the Fano shape of the resonance[45]. The inset in Fig. 5e, shows the reflection intensity as a function of the bias voltage when the input light wavelength is 1486.5nm. From − 12V to + 12V, the reflection amplitude is gradually increasing.

The AC modulation characteristic is tested with the devices in Fig. 5a and plotted in Fig. 6a. A sine wave with peak-to-peak value, $V_{pp}$, of 20V is applied into the device while the wavelength of the incident light is set at 1490nm where the device achieves the highest modulation depth(estimated as 6.8% from Fig. 5a). The sine wave

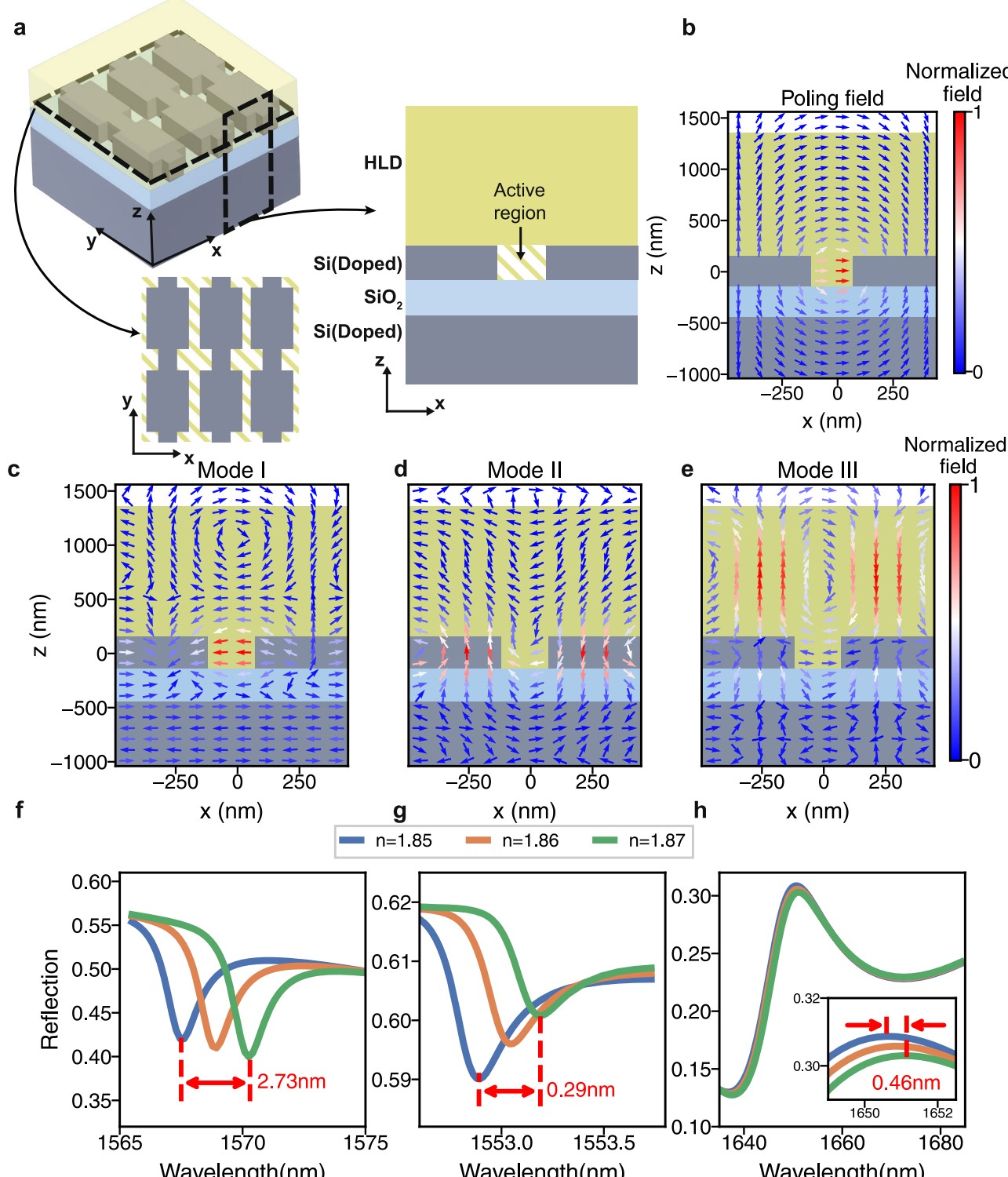

**Fig. 2 | The advantage of slot mode resonance in organic electro-optic modulators. a** The schematic view (top left), top view (bottom left) and cross-section (right) of the device that supports the slot mode. In the schematic view, the OEO material is plotted transparent to show the slot structure underneath. The slots are formed in the device layer of the silicon-on-insulator (SOI) substrate, which is covered by the OEO material HLD. To show the essence of the problem, only the slot is considered as the active region. The dashed rectangles in the schematic view represent the top view across the device layer and the cross-section of a period cell.

**b** The poling field profile when the left and right silicon rail have bias voltages V(V > 0) and 0, respectively. **c**–**e** Normalized electric field profiles for three optical modes that could couple to $E_x$ incident light. **c** the slot mode. **d** the guided mode in the silicon bar. **e** the guided mode in the OEO material. **f**–**h**. the tuning performance of the three optical modes. Figures **f**, **g**, and **h** match with the field profile in figures **c**,**d**, and **e**, respectively. The inset in **h** is a zoom-in spectrum between 1649 nm and 1652.5 nm.

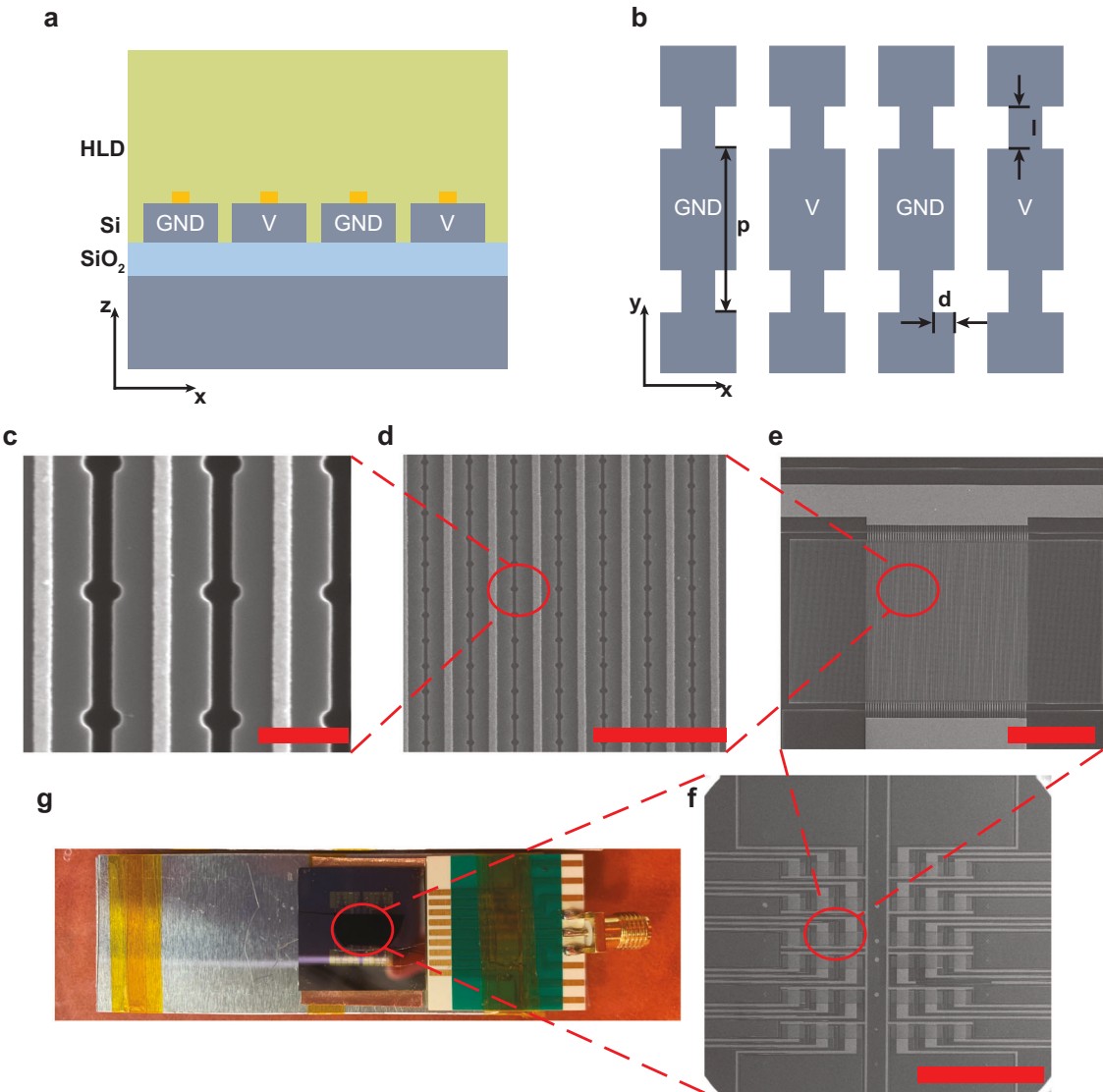

**Fig. 3 | The electro-optic free-space modulator. a–b** The cross-section and top view of the experimentally fabricated device. **c–g** The step-by-step zoom-out image of the device and setup. **c–f** are the SEM images. The scale bars are 500 nm, 3 μm, 50 μm, and 1 mm, respectively. **g** is the optical image of the device. Multiple devices are fabricated within a chip, and they are wire-bonded to the printed circuit board for parallel testing.

frequency is swept from 50kHz to 5.8MHz. The cutoff 3dB bandwidth is at 3MHz. The insets in Fig. 6a show the normalized modulation signal when driving with frequency $f = 80$kHz and $f = 2.8$MHz, respectively. To investigate the AC response, we use a simplified model to predict the AC response shown in Fig. 6b. The model collectively considers the contributing factors to the response speed within and outside the devices. In the devices, we model the nano-bars as a resistance, including the contribution from the gold strip $R_{Au}$ and the silicon nano-bar $R_{Si}$ (see supplementary section 4). The slot is modeled as the parallel connected resistance $R_{OEO}$ and capacitance $C_{OEO}$. Outside the devices, we assume that the major electrical components are the stray capacitance in the circuits, which are split into the capacitance due to the SOI wafer[46] $C_{SOI}$ and other factors $C_{load}$. The parameters in the model are determined by both the geometry of the devices and the AC response of similar devices with different substrates or electrode layouts (See supplementary section 4). The AC response from the prediction of the model is shown as the green line in Fig. 6a, which agrees with the experimental result. Based on the model estimation, the capacitance from the silicon-organic platform and the stray capacitance along the whole circuits are the main factors

preventing the speed increase. The capacitance from the SOI wafer could be solved by advanced CMOS technology used in integrated electro-optic modulators[47]. Judicious material and structural engineering of these circuits have already achieved gigahertz operation of electro-optic modulators[31,48–50]. As a result, there is no fundamental limit in increasing the operation speed up to GHz in our platform.

## Discussion

In this work, we propose a silicon-organic metasurface for free-space modulation with reduced operation voltage less than ±17V. We experimentally observed the resonance with quality factor 330–1310 and up to ~5.5nm shift with nonlinear coefficient $r_{33} = 45.7$pm/V at 1495 nm. The proposed slot mode combines the advantages of a short distance between two electrodes and a large overlap with the OEO material, achieving a tuning sensitivity $S_{abs} = 0.161$nm/V, which shows an improvement with a factor of 1.6 in sensitivity compared to the state-of-art[31,44]. Finally, the metasurface has up to 3MHz bandwidth. The use of the slot mode is not limited to electro-optic systems. The proposed design approach can be applied to any system where sensitivity to perturbations in low-index media is critical. For example, in

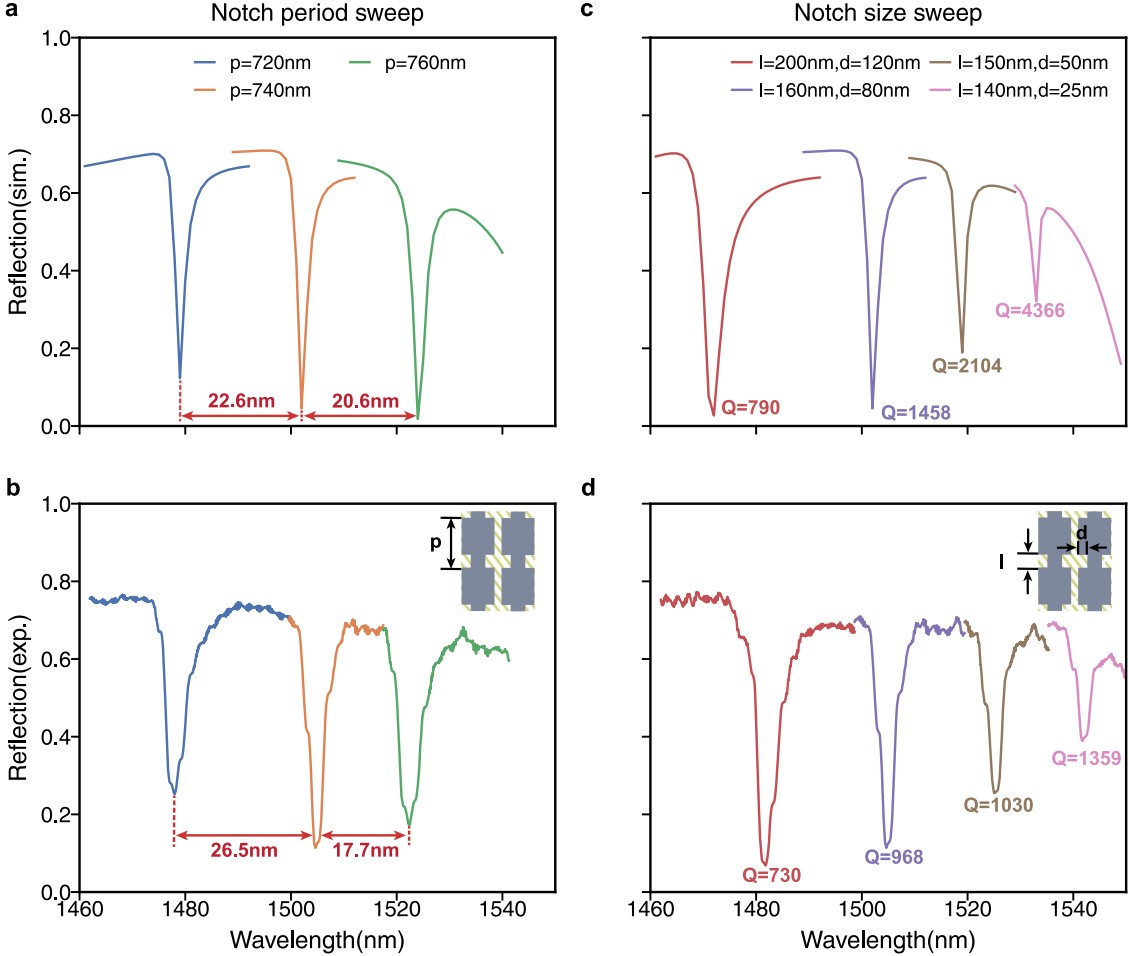

**Fig. 4 | Slot mode resonance characterization. a–d** The simulated (**a, c**) and experimentally measured (**b, d**) reflection spectra when sweeping different sets of perturbation parameters. **a–b** Sweep the periodicity of the notches. All curves have the same notch size $l = 160$ nm, $d = 80$ nm. Blue: $p = 720$ nm. Orange: $p = 740$ nm. Green: $p = 760$nm. The resonance shifts due to periodicity changes are labelled in experiment and simulation curves. c-d. Sweep the notch sizes. All curves have the same notch periodicity $p = 740$ nm. Red: $l = 200$ nm, $d = 120$nm. Purple: $l = 160$ nm, $d = 80$ nm. Brown: $l = 150$ nm, $d = 50$ nm. Pink: $l = 140$ nm, $d = 25$ nm. The quality factor of the resonances are labelled for experimental and simulated plots.

NEMS systems, the slot mode resonance could potentially improve sensitivity to the mechanical movement, compared to conventional guided mode resonances[51].

The currently demonstrated sensitivity is limited by the quality factor and the nonlinear coefficient $r_{33}$. The quality factor could be improved by using smaller notches or a refined fabrication process. The relatively low nonlinear coefficient compared to what we expected[27] is partly due to the surface state of doped silicon used as the electrode[52] and the small width of the slot[41]. Barrier layer protection[53] has the potential to increase the nonlinear coefficient $r_{33}$ by 4-5 times[54], reducing the tuning voltage down to CMOS-level. Judicious doping level adjustments and the microwave coplanar waveguide design could enable GHz speed operation[31,36]. Therefore, with the increase of electro-optic coefficient and operation bandwidth, our platform is a potential solution for GHz free-space modulation at the CMOS level voltage.

This study primarily focuses on structures which are periodic in both $x$ and $y$ dimensions. However, it is not a necessary condition for preserving the slot mode. As a mode propagating along the $y$ direction, the slot mode doesn't necessitate periodic conditions in $x$ direction[55]. By varying the geometry of the slots, individual slot modulation could be potentially achieved. In the $y$-dimension, by introducing high contrast index variations or photonic crystal mirrors, the footprint of the resonant device could be reduced considerably[56,57]. Furthermore, the

proposed devices expect to achieve phase modulation if the over-coupling condition is satisfied by the out-of-plane asymmetry in nanostructures[10] or a bottom mirror[5,6].

In summary, this report presents a low-voltage amplitude modulator using a silicon-organic platform. The slot mode metasurface has the potential to enable high-speed and low-voltage optical switching, sensing, and tuning, for numerous applications such as LiFi, LiDAR, spatial light modulators, and quantum optical communication.

## Methods
### Fabrication and poling methods
The device is fabricated from an SOI wafer, which consists of 300nm p-doped silicon ($1 - 10\Omega \cdot cm$), 300nm BOX (buffered silicon oxide), and $500\mu m$ silicon substrate. The detailed fabrication workflow is shown in supplementary section 6. The device requires two sequential nanofabrication steps for the silicon rails and metallic strips, respectively. Both E-beam lithography steps utilize ZEP-520A (Zion Corporation) as the resist, 100kV electron beam (EBPG-5200, Raith GmbH) to expose, and ZED-N50 (Zion Corporation) as the developer. After the first E-beam lithography, we use the resist as the soft mask and the pattern is transferred to silicon by ICP-RIE etching (PlasmaLab System 100, Oxford Instrument). Next, the resist is removed by Remover PG. The second E-beam lithography writes the liftoff mask for the electrodes, following which 5nm Ti

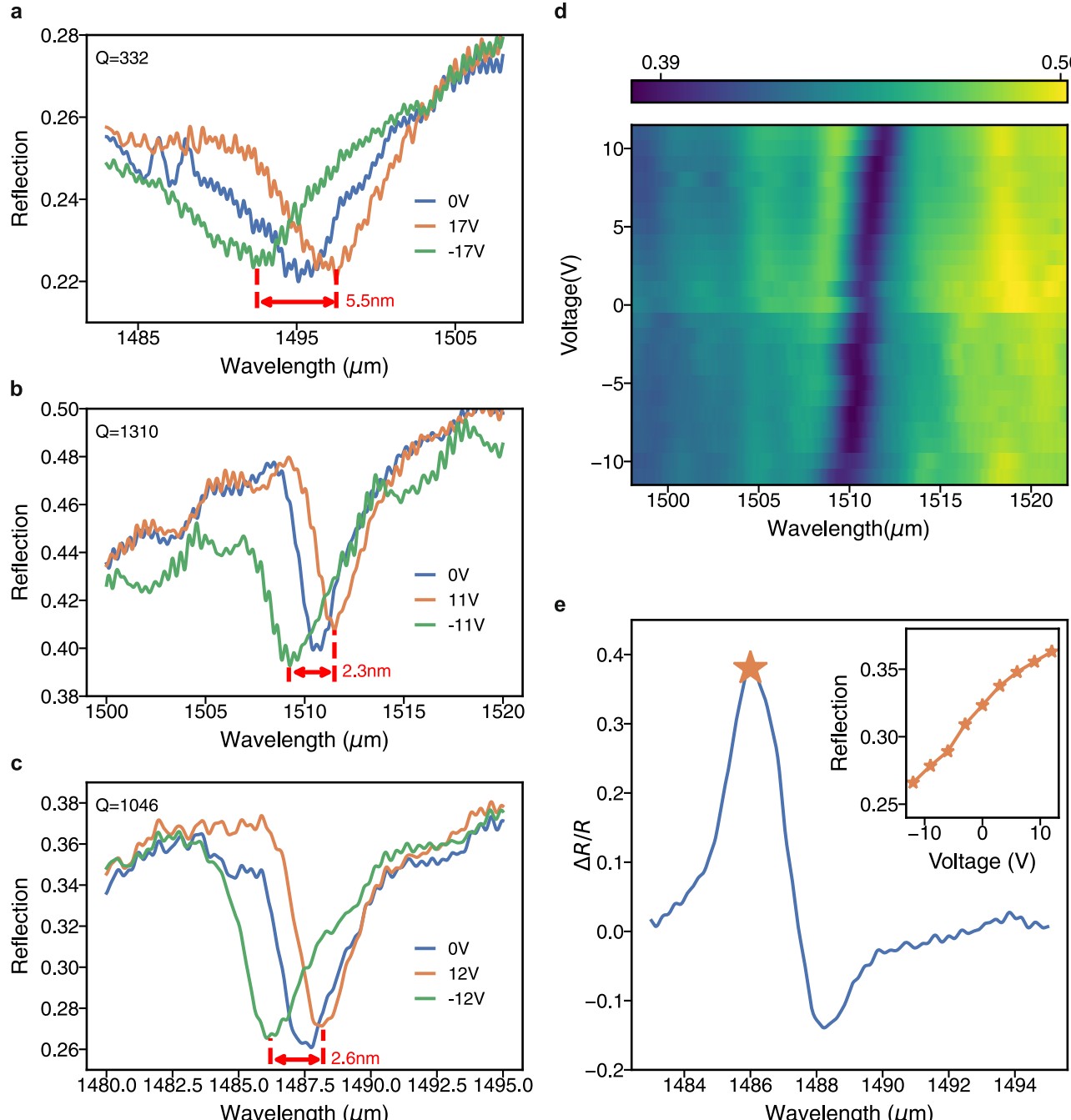

**Fig. 5 | DC tuning characteristics. a–c** Reflection spectra of three different devices under DC tuning. The applied biases are denoted in the legend. **d** The reflection spectra of the device in **b** with bias voltages ranging from -11V to 11V. **e** The maximum modulation ratio ($\Delta R/R = (R_{max} - R_{min})/R_{V=0}$) for each wavelength in device shown in **c**. The inset depicts the absolute reflection as the DC bias voltage is swept from -12V to 12V for a fixed wavelength of incident light of 1486nm. The absolute reflection changes over 10%.

and 60nm Au are deposited sequentially using an E-beam evaporator (Kurt J. Lesker E-beam evaporator). Liftoff is then performed in Remover PG. Finally, a layer of OEO material (HLD, NLM Photonics) is spin-coated on top of the device, followed by a 3-hr solvent removal in a vacuum oven at 65° C. The detailed workflow is shown in supplementary section 6.

Poling of the HLD material is performed by heating the device under the nitrogen environment while applying a poling voltage. This voltage creates a poling field around 100V/$\mu$m across the slot. The heating process consists of a 6° C/s temperature ramping, 5 to 10 minutes of holding at 95°C, and rapid cooling.

## Simulation methods

The simulations in Fig. 2, and Figs. S1–S3 use COMSOL Multiphysics software. The periodic condition is applied in both x and y directions. Refractive indices of the silicon, silicon oxide, and OEO material are assumed to be 3.52, 1.44, and 1.85. Figure 4a, c are simulated with a FDTD simulation software (Ansys Lumerical FDTD) with periodic boundary conditions applied in x and y.

## Measurement

The measurements were conducted using the experimental setup depicted in Fig. S7. The light source utilized was a tunable external-cavity

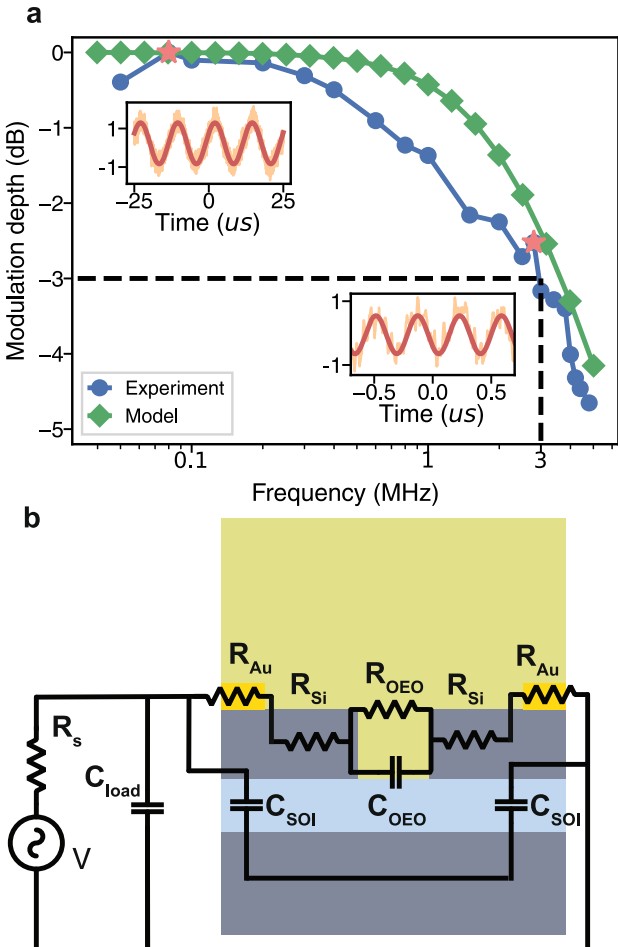

**Fig. 6 | AC tuning characteristics and circuit model scheme. a** The experimental and modeled AC response of the active device. Blue curve: the experimental normalized modulation depth. The cutoff bandwidth is at 3MHz. Green curve: the model prediction of the modulation depth. Insets: the example modulation signal (Orange) and the fitting sine wave (Red) in the time domain. The frequencies of the modulation signal are 80kHz (top left) and 2.8MHz (bottom right), respectively. **b** The circuit model of the device. $V$: the voltage source. $R_s$: the source resistance. $C_{load}$: external stray capacitance outside the device. $C_{SOI}$: the capacitance due to the use of SOI wafer. $R_{Au}$: the resistance of the gold strip on top of the silicon nano-bar. $R_{Si}$: the resistance of the silicon nano-bar. $C_{Poly}$: the OEO material capacitance. $R_{Poly}$: the OEO material resistance.

diode laser (Toptica CTL-1550). A fiber collimator (Thorlabs, F260FC-1550) was employed to collimate the beam. A beam splitter is placed in front of the collimator to split a small amount of power for the reference InGaAs detector (Thorlabs, PDA10CS). The polarized state of the incident light was set to TE polarization using a linear polarizer. Then the light goes through a beam splitter and only half of the power is used.

A × 20 infinity-corrected objective lens (Mitutoyo, M Plan Apo NIR) and a tube lens with a focal length of 200mm were used to image the sample at the object plane, with the tube lens and the mounting stage of the sample adjusted to ensure normal incidence. An iris (Thorlabs, ID25) was inserted at the image plane to select a region of interest with a diameter of $45\mu m$ in the object plane. The spatially filtered light was either focused onto another InGaAs detector for the measurement of the spectra or imaged on an InGaAs SWIR camera (Goodrich, SU320HX-1.7RT) using relay optics. The reflection signals were obtained by dividing the signal from the sample by the signal from the sources. To reduce the incident power variation at different wavelengths, the signals were further normalized by the signals from the exposed gold pad on the chip.

## Data availability
The data that support the findings of this study are available from the corresponding author upon request.

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

## Acknowledgements

We thank NLM Photonics for the HLD OEO material and for consulting on the preparation and poling process of the HLD. The device nanofabrication was performed at the Kavli Nanoscience Institute at Caltech. This work was supported by the Caltech Sensing to Intelligence program and ARO W911NF2210097. H.K. acknowledges a fellowship from Ilju organization.

## Author contributions

H.K., T.Z. and A.F. conceived the project. A.F. supervised the project. H.K and T.Z. designed the structures. T.Z. and Y.G. fabricated devices, performed simulations and measurements, and analyzed data. T.Z. designed and prepared the printed circuit boards. G.R. provided feedback on the design of the structures. T.Z. wrote the manuscript. All authors discussed the results and commented on the manuscript.

## Competing interests

The authors declare the existence of the following financial competing interest: T.Z, Y.G, H.K., and A.F. have filed for a patent application (number: 18/189074) based on the results of this paper. There are no other competing interests.
