## [Peer Review File · Nature Communications]

Dynamic Light Manipulation via Silicon-Organic Slot MetasurfacesREVIEWER COMMENTS

Reviewer #1 (Remarks to the Author):

This paper demonstrates a device for free space beam manipulation (spatio-temporal control) based on an organic electro-optic material in a metamaterial surface with a Megahertz frequency response and +/-10V power requirements.

The concept is original, the device performance over prior art (Refs. 26, 28) is better. Conceptually, the authors use highly doped silicon slot waveguides into the metamaterial surface to enhance the electro-optical efficiency. This way the authors reduced the required voltage over prior art published results on active metamaterial modulated surfaces quite a bit. Voltages in the order of 10V rather than 100V are used. Still, there is room for improvement. The modulation depth at best is 40%.

Generally, the paper is very well written. However, if it is to be published then the authors should revise the following points (which are listed not in order of relevance).

a) The title claims "low-voltage dynamic light manipulation with silicon-organic slot metasurfaces". Since >20 Vpp was needed to modulate the reflection by <1.5 dB, the low voltage statement should be used carefully. "Reduced voltage requirement" might be better. Low voltage is in the order of 1 V. See e.g. the liquid crystal (and of course much slower) work here

<https://doi.org/10.1002/adv.202102646>

<https://doi.org/10.1038/s41467-022-34494-0>

The authors might claim MHz though.

b) In the abstract: The given metric of tuning the resonance wavelength by nm/V is interesting but there are other relevant metrics such as the extinction ratio, the 3-dB bandwidth, voltage amplitude that should be stated.

c) If the authors want to hold on to the CMOS-level voltages in the abstract, then they should show a path on how to come from the 10V to the 1 V level.

d) Just above (1): According to the eq. (1), the refractive index varies in time at certain locations. It does not change in space.

e) A conductivity of $1\text{-}10\ \Omega\cdot\text{cm}$ is claimed, corresponding to $1\text{e}15\text{-}1\text{e}16$ dopants/cm³. Later the same silicon is called "highly doped". Since typically, highly doped means $\sim 1\text{e}19$ dopants/cm³ I would remove the qualifier and just call it "doped" silicon.

f) The AC experiment should state the achieved modulation depth explicitly.

g) The bandwidth of 3 MHz is measured directly. A lumped-element model is assumed according to Fig. 6. b. The authors should indicate measured total capacitance and resistance and the corresponding RC-bandwidth, since they claim the possibility for GHz modulation. I.e. 300x bandwidth improvement. They should further elaborate, how they plan to avoid the RC-limitation. They currently state that "judicious doping level adjustments and the microwave coplanar waveguide design" could enable GHz speed operation.

Using the model provided in Fig. 6.b. and the parameters estimated in the supplement:

$R_{Au} = 338\ \Omega$, $R_{Si} = 1.17\text{M}\Omega$, $R_S = 50\ \Omega$, $COEO = 21.25\text{fF}$. An RC-bandwidth of

$1/(2\pi \cdot (R_{Au} + R_{Si} + R_S) \cdot COEO) = 6.4\ \text{MHz}$ can be calculated. An increased doping can reasonably reduce R_{Si} by 300x and therefore increase the RC-bandwidth towards $\sim 1\text{GHz}$. However, the impact on optical loss, therefore Q-factor and device efficiency needs to be addressed.

The authors cite references 28 and 32 to justify their claim. However, reference 32 utilizes wide signal and ground lines of 15 μm and 80 μm respectively and an additional 4 μm gap between signal and ground to achieve impedance matching. This would result in an active area of just 0.05% of the total device area, which will not be efficient. The authors should remove this reference or provide a reasonable explanation how they want to increase the bandwidth, retaining device efficiency. Reference 28 utilizes metal electrodes, avoiding the doped silicon, which in this work is integral part of the device. It is not clear, how the authors could apply the principles from reference 28 and increase the bandwidth. The authors should remove the reference or provide a reasonable explanation how they want to increase the bandwidth.

h) The experimental setup is not depicted in Fig. S5. This should probably refer to Fig. S7

i) "the signals were further normalized by the signals from the gold"-> which gold?

- j) Figure 3 g. clearly shows a metal behind the chip. Since the silicon substrate behind the device is transparent in the relevant wavelength, there is inevitably specular reflection interfering with the measured signal. How did the authors subtract this contribution from the reflection of the device? It is unclear, whether this reflection is part of the described device.
- k) How large is the spot size in relation to the active area?
- l) Equation (S1) cannot be correct as according to (1) Γ is a scalar without units and the term on the right side of (S1) has units of V^2/m^2 .
- m) Same for (S2).
- n) In the main text, the overlap factor is called Γ , here it is inconsistently denoted Γ_c .
- o) Equation (S1) cannot be found in neither reference 4, nor in reference 5.

Reviewer #2 (Remarks to the Author):

In the submission, Zheng et al. proposed and experimentally demonstrated a tunable metasurface by coupling a slot-mode metasurface with an organic material. By carefully engineering the metasurface design to minimize the electrode separation and maximize the mode overlap, a relatively low-voltage operation and large modulation amplitude is achieved. The paper is well-written and may be of high interest to the metasurface research community. However, before making a final recommendation, I still have the following concerns and suggestions:

1. The modulation speed of the current device is not particularly high. In addition, this is achieved with a small aperture size of $\sim 80 \times 100 \text{ um}^2$. For a practical device with a larger aperture size, the RC constant may further increase. I am concerned this may be a large limiting factor for applications like LiFi. Is there a potential solution?
2. For the proposed LiFi application, is there any significant advantage to having an external light intensity modulator compared to the direct modulation of the LED?
3. For phase modulation and applications like LiDAR, it is important to achieve wide-range phase modulation. In addition, for the quasi-BIC mode used, the coupling between neighboring unit cells may pose a major issue for beam forming. The authors claimed this may not be an issue by citing other papers that are not directly related. I feel it may be necessary to at least do a numerical simulation to verify this point.
4. In the introduction, I suggest the authors at least discuss the approach based on liquid crystal. This is the approach that seems most promising for near-term applications (www.lumotive.com; US Patent 10,451,800; US Patent 10,665,953).
5. How to ensure the complete filling of the organic material inside the slot? From previous personal experience, this is not trivial. Bubbles may form in the gap, which may significantly deteriorate the device's performance. This may be verified by taking a cross-sectional image of the device using FIB.

Re. Zheng et al. Low-voltage dynamic light manipulation with silicon-organic slot metasurfaces

Reply to reviewers' comments

10/18/2023

We greatly appreciate the reviewers for their thorough evaluation of our manuscript and for recognizing the novelty of our work. In our following responses, we address the constructive comments raised by the reviewers and hope that our comprehensive explanation regarding our designs and experiments will help alleviate the concerns of both reviewers. Our work represents another innovative finding in the field of active metasurfaces as it demonstrates the potential of slot modes that are very sensitive to perturbations, for applications in amplitude and phase modulation.

We have incorporated our point-by-point responses to the reviewers' comments and indicated the corresponding changes made in the manuscript. For the improvement of readability, our responses are printed in blue font, while the corresponding sentences in the manuscript are displayed in green with the modified parts in red. We also note that all references cited in this reply can be found at the end of this document.

Reviewers Comments:

Reviewer #1 (Comments for the Author):

This paper demonstrates a device for free space beam manipulation (spatio-temporal control) based on an organic electro-optic material in a metamaterial surface with a Megahertz frequency response and +/-10V power requirements.

The concept is original, the device performance over prior art (Refs. 26, 28) is better.

Conceptually, the authors use highly doped silicon slot waveguides into the metamaterial surface to enhance the electro-optical efficiency. This way the authors reduced the required voltage over prior art published results on active metamaterial modulated surfaces quite a bit.

Voltages in the order of 10V rather than 100V are used. Still, there is room for improvement.

The modulation depth at best is 40%.

Generally, the paper is very well written. However, if it is to be published then the authors should revise the following points (which are listed not in order of relevance).

Our response: We are grateful for the reviewer's concise assessment of the manuscript and the encouraging remark on the work. The questions will be addressed point by point.

- a) *The title claims "low-voltage dynamic light manipulation with silicon-organic slot metasurfaces". Since >20 Vpp was needed to modulate the reflection by <1.5 dB, the low voltage statement should be used carefully. "Reduced voltage requirement" might be better. Low voltage is in the order of 1 V. See e.g. the liquid crystal (and of course much slower) work here. <https://doi.org/10.1002/adv.202102646> <https://doi.org/10.1038/s41467-022-34494-0> The authors might claim MHz though.*

Our response: We appreciate the reviewer's insightful feedback regarding the title. We concur that, although the slot waveguides reduce the voltage requirement for electro-optic tuning, the voltage requirement of slot waveguides is still higher than liquid crystals. We intended the term 'low-voltage' as a relative term, in reference to previous works on free-space tuning that utilized organic electro-optic materials, which required significantly higher voltage. To ensure there is no ambiguity or misunderstanding, we have decided to remove the term but describe the advantage of this platform in the abstract and main texts. The revised title will now read: 'Dynamic Light Manipulation via Silicon-Organic Slot Metasurfaces'.

- b) *In the abstract: The given metric of tuning the resonance wavelength by nm/V is interesting but there are other relevant metrics such as the extinction ratio, the 3-dB bandwidth, voltage amplitude that should be stated.*

Our response: We are grateful for the reviewer's suggestion to include relevant performance metrics in the abstract. Following this guidance, we have incorporated the suggested details in the manuscript:

We demonstrate the maximum tuning sensitivity of 0.16nm/V, the maximum extinction ratio of 38% within $\pm 17V$ voltage at telecommunication wavelength. The device has a 3dB bandwidth of 3MHz.

- c) *If the authors want to hold on to the CMOS-level voltages in the abstract, then they should show a path on how to come from the 10V to the 1 V level.*

Our response: As outlined in the main body of our manuscript, we achieved a Pockels coefficient of 45.7pm/V in our experiment. This value is approximately ten times less than the reported value for bulk organics in reference 24[1], and about four times less than that for in-slot organics in reference 43[2]. We also believe that the Pockels coefficient could be substantially improved through strategic choices in organic molecule selection and by optimizing the polling process. As demonstrated by Equation 2 in our manuscript, the tunability exhibits a linear relationship with both the Pockels coefficient (r_{33}) and the applied voltage. Therefore, if we were to increase the Pockels coefficient to a value that is four times greater than our current achievement, the operating voltages could feasibly be reduced to the CMOS-level, for instance, between 0-5V. To clearly show the path to achieve CMOS-level voltage, we add the following texts in the discussion section of the main manuscript:

The relatively low nonlinear coefficient compared to what we expected is partly due to the surface state of doped silicon used as the electrode and the small width of the slot. Barrier layer protection has the potential to increase the nonlinear coefficient r_{33} by 4-5 times, reducing the tuning voltage down to CMOS-level.

- d) *Just above (1): According to the eq. (1), the refractive index varies in time at certain locations. It does not change in space.*

Our response: We are grateful to the reviewers for highlighting the potentially confusing description in our manuscript. When considering the precise formula for the resonance shift, which includes the overlapping factors, the change in the refractive

index must be considered as both tensorial and spatially variant (as detailed in Supplementary Section 2). However, to maintain the readability and focus of the main text, we have opted to avoid excessive mathematical details which could potentially distract readers from comprehending the core insight of our structural design process. Consequently, we have revised the corresponding text accordingly:

When the operating signal is applied, the active layer induces a refractive index change

$$\Delta n(t) = -1/2 n_{mat}^3 r_{33} E_{ext}(t) \quad (1)$$

where E_{ext} is the external electric field in the OEO material. Besides the operating voltage or the field in slots, the perturbation strength to the optical mode also depends on the overlap factor Γ_c between the electric field profile and the optical mode profile. Γ_c is irrelevant to the external voltages. To calculate this overlap factor, we need to treat E_{ext} , r_{33} as spatially dependent variables. Based on cavity perturbation theory (See the full derivation in supplementary section 2), we could extract the formula to calculate the shift of the resonant frequency.

- e) A conductivity of 1-10 $\Omega \cdot \text{cm}$ is claimed, corresponding to 1e15-1e16 dopants/cm³. Later the same silicon is called "highly doped". Since typically, highly doped means ~1e19 dopants/cm³ I would remove the qualifier and just call it "doped" silicon.

Our response: We are appreciative of the reviewers' comments concerning the doping level of our device. We acknowledge that a range of 1e-15 to 1e16 does not constitute a particularly high doping level and, as such, we will refrain from using the term 'highly doped'. We have edited the subsequent sentences in our manuscript accordingly to reflect this:

The relatively low nonlinear coefficient compared to what we expected is partly due to the surface state of doped silicon used as the electrode and the small width of the slot.

- f) The AC experiment should state the achieved modulation depth explicitly.

Our response: We are grateful for the reviewer's astute observation regarding the omitted condition in our AC experiment. Indeed, we conducted our AC experiment using the chip depicted in Fig. 5a. Based on our DC tuning experiment, we recorded a modulation depth of 12.8%. The modulation depth of the device at 17V is illustrated in Fig. R1, which effectively sets the upper limit for our AC modulation depth.

However, during the AC experiment, we only applied a maximum voltage of $\pm 10V$. This reduced voltage was intentional to avoid the potential for breakdown at high speed. Regrettably, we did not directly measure the reflection spectra at $\pm 10V$. Instead, we will infer the modulation depth using the measured data from the DC experiment. The shift of the $\pm 17V$ voltages is 5.48nm, which gives the shift 0.1611nm/V. We manually shift the reflection spectra without bias for $\pm 1.61nm$, and calculate the estimated modulation depth ($\Delta R/R = (R_{10V} - R_{-10V})/R_{0V}$) in Fig. R2:

Fig. R2. Left: the inferred reflection spectra at tuning voltage 10V and -10V. The shift of the spectrum is estimated by the measurement of the shift under the voltage 17V and -17V. Right: the inferred modulation depth when applying 10V/-10V.

At wavelength 1490nm, the modulation depth is 6.8%. Therefore, we have added the modulation depth of the device used in Fig. 6:

A sine wave with peak-to-peak value, V_{pp} , of 20 V is applied into the device while the wavelength of the incident light is set at 1490 nm where the device achieves the highest modulation depth (estimated as 6.8% from Fig. 5a).

- g) *The bandwidth of 3 MHz is measured directly. A lumped-element model is assumed according to Fig. 6. b. The authors should indicate measured total capacitance and resistance and the corresponding RC-bandwidth, since they claim the possibility for GHz modulation. I.e. 300x bandwidth improvement. They should further elaborate, how they plan to avoid the RC-limitation. They currently state that “judicious doping level adjustments and the microwave coplanar waveguide design” could enable GHz speed operation.*

Using the model provided in Fig. 6.b. and the parameters estimated in the supplement: $R_{Au} = 338\Omega$, $R_{Si} = 1.17M\Omega$, $R_S = 50\Omega$, $C_{EO} = 21.25fF$. An RC-bandwidth of $1/(2\pi \cdot (R_{Au} + R_{Si} + R_S) \cdot C_{EO}) = 6.4$ MHz can be calculated. An increased doping can reasonably reduce R_{Si} by 300x and therefore increase the RC-bandwidth towards ~1GHz. However, the impact on optical loss, therefore Q-factor and device efficiency needs to be addressed.

The authors cite references 28 and 32 to justify their claim. However, reference 32 utilizes wide signal and ground lines of 15 μm and 80 μm respectively and an additional 4 μm gap between signal and ground to achieve impedance matching. This would result in an active area of just 0.05% of the total device area, which will not be efficient. The authors should remove this reference or provide a reasonable explanation how they want to increase the bandwidth, retaining device efficiency. Reference 28 utilizes metal electrodes, avoiding the doped silicon, which in this work is integral part of the device. It is not clear, how the authors could apply the principles from reference 28 and increase the bandwidth. The authors should remove the reference or provide a reasonable explanation how they want to increase the bandwidth.

Our response: We appreciate the reviewer's inquiry regarding the potential enhancement of the bandwidth. Given the complexity and depth of this question, we have decided to provide a response in three separate parts:

1. The reviewer asked about the direct measurement of the RC constant in our circuit and the detailed way we propose to achieve the GHz speed.

Given that our sample is largely coated with polymer, we only have access to the voltage response on the electrode pad shown in Fig. S5. Due to repeated wire bonding, the electrodes have incurred substantial damage. These factors make the direct measurement arduous. Therefore, we have conducted a series of bandwidth measurements, detailed in Supplementary Section 4, to identify the primary factor limiting speed. Based on the speed tests conducted in Experiments I, II, and III, we are able to approximate the capacitances and resistances within our main circuit. Our calculations indicate that the external capacitance (C_{load}) and the silicon-on-insulator capacitance (C_{SOI}) are the

principal factors restricting bandwidth. Consequently, we suggest the following strategies for improving the bandwidth, potentially even up to the GHz range:

- i) Use doped silicon on fused silica wafer to remove C_{SOI} . This type of wafer will require the development of a specialized fabrication process, but it could be potentially constructed by ion implantation and annealing. This improvement could be supported by experiment I and II, as the replacement of the wafer will lead to the increase of the bandwidth in experiment I.
 - ii) The design and implementation of a microwave coplanar waveguide, combined with the use of a probe instead of wire bonding and meticulous removal of possible parasitic capacitance in the circuit, could effectively eliminate C_{load} , and it has been shown in reference 28. We expect that in ideal case the bandwidth should only be restricted by R_{Au}, R_{Si} and C_{OEO} (R_{OEO} is ignored due to the high resistance) .
 - iii) The careful adjustment of doping levels, device length, and device geometry to tailor them toward achieving GHz bandwidth presents another feasible avenue for optimization. We will further elaborate on the particulars of this approach in the subsequent part of our response.
2. The reviewer calculated the main restriction bandwidth and asked about the way of improvement.

First, we would like to extend our gratitude to the reviewer for their meticulous calculations. We concur that the principal limitation of the bandwidth is indeed the RC constant of the nanobars. Upon careful examination, we have identified a typographical error in our supplementary material where R_{Si} was mistakenly listed as $1.17M\Omega$ instead of the correct value of $1.17k\Omega$. We sincerely apologize for this oversight. With the corrected value, the RC-bandwidth is actually 4.8GHz rather than the previously stated 6.4MHz. Consequently, we believe that an aggressive 300-fold increase in doping is unnecessary to achieve GHz speeds. The aforementioned modifications, along with the detailed strategies in our response, lead us to be optimistic about reaching our targeted bandwidth.

3. The reviewer asked about the role of reference 28 and 32.

The references 32 and 28 serve for distinct purposes for our work. Reference 32 features a 100GHz organic EO modulator, illustrating that the speed bottleneck is not constrained by the electro-optic effect response speed. Theoretically, if resistance and capacitance along the circuit could be minimized, speeds up to 100GHz may be attainable. However, as the reviewer astutely pointed out, achieving this speed would necessitate a small fill factor of the device area, rendering such speeds unfeasible in our particular scenario.

Conversely, reference 28 showcases a GHz free-space modulator, with a 3dB performance at 3 GHz. This reference provides insight into the possibility of eliminating our additional capacitances C_{load} and C_{SOI} . Furthermore, by contrasting the RC constant of reference 28 with that of our device, we can

delineate the potential upper limit of the bandwidth that may be attainable with our doped silicon device.

In reference 28, the estimated resistance of the whole device is $R_{Au,w} = 24\Omega$, and its capacitance is $C_{OEO,w} = 0.27pF$. Considering their number of pairs $N_z = 240$, and the bar length is $300\mu m$ while our device has only $100\mu m$, the 'normalized' resistance and capacitance of a single pair are $R_{Au,s} = R_{Au,w} * N_z/3 = 1920\Omega$ and $C_{OEO,s} = C_{OEO,w}/N_z/3 = 0.375fF$. Therefore, the bandwidth of the metal device in reference 28 could be estimated to be $1/(2\pi R_{Au,s} C_{OEO,s}) = 23.4GHz$ if the device footprint is only $100\mu m$.

Comparing these values with our device, the capacitance $C_{OEO} = 21.25fF$ is much higher, and the resistance $2(R_{Au} + R_{Si}) = 3016\Omega$ remains similar. Therefore, the GHz bandwidth is potentially attainable. It should be emphasized that, although the doped silicon has resistivity that is $\sim 10^7$ times compared to gold resistivity, the gold nanobar's presence on top of the silicon will greatly reduce the silicon resistance to the similar level of the gold since the nanobar is $\sim 100\mu m$ long but only $700nm$ wide. To facilitate this observation, we conducted the COMSOL simulations to achieve the resistance of the single pair nanobars. The geometry settings are shown in Fig. R3. The first simulation shorted the entire circuit by moving the two nanobars together, yielding a device resistance of 3639.1Ω . If we remove the silicon nanobar and directly move the two gold nanobars together, the resistance is 234.5Ω . That is the similar order of magnitude compared to our approximation. Therefore, we expect that even without any doping level increase, there is no limitation of the GHz bandwidth in our device.

In conclusion, although our current device shows the bandwidth of 3MHz, we expect that replacement of the substrate and improvement on the circuit could lead to GHz bandwidth. We have added the detailed path towards GHz bandwidth in the supplementary section 5. We believe this comprehensive analysis addresses the

Fig. R3. The top view of the geometry setting in numerical simulation for R_{Si} and R_{Au} . Left: the geometry of numerical simulation for R_{Si} . $R_{Si} = 3639.1\Omega$. Right: the geometry of numerical simulation for R_{Au} . $R_{Au} = 234.5\Omega$. The geometry parameters are the same as the device shown in Fig. 5a in main text.

reviewer's concerns and offers a robust argument for the feasibility of our proposed improvements.

- h) *The experimental setup is not depicted in Fig. S5. This should probably refer to Fig. S7 “the signals were further normalized by the signals from the gold”-> which gold?*

Our response: We are grateful for the reviewer’s for correcting the typo and requesting clarification. The figure of the experiment setup is S7. The sentence “*the signals were further normalized by the signals from the gold*” signifies that we also measure the signal from an area that has gold coated on top. We assume that the reflection from the gold area is 100% and use it for normalization. (Due to this normalization we expect the real reflection is a few percent larger). To further elucidate this procedure, we have revised the statement:

Thus, the signals were further normalized by the signals from the exposed gold pad on the chip.

- i) *Figure 3 g. clearly shows a metal behind the chip. Since the silicon substrate behind the device is transparent in the relevant wavelength, there is inevitably specular reflection interfering with the measured signal. How did the authors subtract this contribution from the reflection of the device? It is unclear, whether this reflection is part of the described device.*

Our response: We appreciate the reviewer's inquiry regarding the detailed measurement procedure. In our experiment, the backside of the chip is affixed with copper tape, which indeed reflects a signal. However, this reflected signal is not collected for several reasons:

1. The silicon-on-insulator (SOI) wafer is **single-side-polished (SSP)**. This single-side polishing causes the backside of the wafer to scatter most of the light.
2. The surface of the copper tape itself is **non-polished**, and we expect that it will also scatter a significant amount of the light.
3. **The substrate is doped**, resulting in some loss during the light propagation. As the substrate thickness is 500um, the absorption further reduces the intensity of the reflected light.

Consequently, the power from the light reflected off the backside of the chip is minimal and can be considered negligible in our analysis.

j) *How large is the spot size in relation to the active area?*

Our response: The laser spot used is a circle with a diameter of 30 μm , while the device itself has an area of 80 μm \times 100 μm (width \times length). During the experiment, the spot is carefully positioned at the center of the device by the SWIR camera. It is important to note that the active area of the device is somewhat smaller than the overall device area. This reduction is due to the edges of the device not maintaining the necessary periodic conditions. However, given that the laser spot size is considerably smaller than the overall device area, we are confident that the active area of the device fully encompasses the spot.

k) *Equation (S1) cannot be correct as according to (1) Γ is a scalar without units and the term on the right side of (S1) has units of V^2/m^2 .*

l) *Same for (S2).*

m) *In the main text, the overlap factor is called Γ , here it is inconsistently denoted Γ_c .*

n) *Equation (S1) cannot be found in neither reference 4, nor in reference 5.*

Our response: We are grateful to the reviewer for identifying the unit discrepancy in Equations (S1) and (S2). Upon careful review, we acknowledge an oversight in the normalization of Equation (S1). This equation has its origins in Equation (24) within the supplementary material of Reference 4. To address this concern, we have revised the discussion of the overlapping factor in **Supplementary Section 2**. Specifically, we have employed perturbation theory to derive the equation below (also the equation (S15) in supplementary material) that aligns with Equation (24) in the supplementary material of Reference 4. This revised derivation ensures the consistency and accuracy of our mathematical treatment.

$$\Gamma_c = \frac{w_s^2 \iiint_{poly} \epsilon |E_{ext} \cdot E_i|^2 dx dy dz}{V_{ext}^2 \iiint_{everywhere} \epsilon |E_i|^2 dx dy dz}$$

V_{ext} is the tuning voltage, w_s is the slot gap width, E_{ext} is the electric field induced by tuning voltage, E_i is the i th optical eigenmode, ϵ is the dielectric constant. Since E_{ext} has a linear relationship of tuning voltage, the overlap doesn't depend on the value of the tuning voltage.

The Γ_c in the supplementary material is the same symbol as the Γ in the main text. We have used Γ_c in all texts.

Reviewer #2:

In the submission, Zheng et al. proposed and experimentally demonstrated a tunable metasurface by coupling a slot-mode metasurface with an organic material. By carefully engineering the metasurface design to minimize the electrode separation and maximize the mode overlap, a relatively low-voltage operation and large modulation amplitude is achieved. The paper is well-written and may be of high interest to the metasurface research community. However, before making a final recommendation, I still have the following concerns and suggestions:

Our response: We are grateful for the reviewer's insightful evaluation of the manuscript and their supportive comments concerning the work. The questions will be addressed point by point.

1. *The modulation speed of the current device is not particularly high. In addition, this is achieved with a small aperture size of $\sim 80 \times 100 \text{ um}^2$. For a practical device with a larger aperture size, the RC constant may further increase. I am concerned this may be a large limiting factor for applications like LiFi. Is there a potential solution?*

Our response: We appreciate the reviewer's insights regarding the speed limitations of our device, particularly as it relates to its size. We agree that a larger device size may result in reduced bandwidth. However, we believe that an effective strategy would involve shrinking the device size to create an individually tunable array. This approach aligns with our intention to leverage the capabilities of 2D spatial light modulators, thereby enabling advanced functionalities, such as those applicable in LiFi[3]. In the subsequent response, we will delve into the specific advantages of utilizing a metasurface as a modulation system in LiFi application.

2. *For the proposed LiFi application, is there any significant advantage to having an external light intensity modulator compared to the direct modulation of the LED?*

Our response: In LiFi systems, LEDs or laser diodes indeed have advantages of modulating intensity with simple integration and low power consumption. However, as mentioned above, the reported devices could provide a different approach by modulating light pixel-by-pixel. Acting as the reconfigurable intelligent surfaces (RIS) in free-space optic (FSO) systems, the reported device array could unlock new possibilities in LiFi applications. For example, a 2D array of phase modulation devices can enable light beam steering, which then can enhance coverage of the LiFi transmitter[3] and establish new links in an environment with obstacles[4]. Additionally, if we incorporate many devices with an angular diversity transmitter (ADT), the amplitude modulation devices could provide less interference compared to fixed structures [3].

3. *For phase modulation and applications like LiDAR, it is important to achieve wide-range phase modulation. In addition, for the quasi-BIC mode used, the coupling between neighboring unit cells may pose a major issue for beam forming. The authors claimed this may not be an issue by citing other papers that are not directly related. I feel it may be necessary to at least do a numerical simulation to verify this point.*

Our response: We are thankful to the reviewer for inquiring about future potential applications, such as LiDAR. We agree that phase modulation is essential to enable this application. One approach to achieving a 2π phase response within the resonance involves the utilization of a back gold mirror, as described in reference [5]. In line with this concept, we propose the introduction of a gold layer beneath the silicon oxide, as depicted in Fig. R4. By meticulously redesigning the geometry of the structure, we anticipate satisfying the over-coupling condition. Our simulations in COMSOL support this idea, with numerical results in Fig. R4 revealing a 1.62π phase response.

Moreover, the proposed platform opens the door to the realization of a 1D spatial light modulator. We acknowledge the reviewer's concern regarding the potential interference from neighboring unit cells, particularly in the context of beam forming. Although the slot mode is primarily confined within the slot waveguide, some leakage of the mode field may cause coupling between adjacent unit cells. As an initial concept to mitigate this challenge, we suggest combining multiple slots into a single pixel to reduce neighboring coupling. Without specific optimization for slot mode confinement, we performed simulations using 3 pixels, with 15 slots in each. These simulations showed a promising first-order diffraction efficiency of 67.6%, as seen in Fig. R5. We believe that further optimization of the slot mode confinement will further reduce the pixel size of the spatial light modulator. We hope that these numerical simulations will clarify our proposed direction and the potential avenues we could explore in our future research.

Fig. R4. Enhanced phase response by backside gold mirror. Left: A cross-sectional view of the proposed phase-enhanced device, with an additional gold mirror incorporated beneath the silicon oxide layer. Right: Simulated reflection intensity and phase, displaying a phase coverage of 1.62π .

Fig. R5. The beam steering performance. Left: A schematic illustration of the simulation's geometry, where we consider N slots as a single pixel and apply a phase gradient across different pixels. Right: The resulting performance when $N=15$, achieving a 67.6% first-order diffraction efficiency.

4. *In the introduction, I suggest the authors at least discuss the approach based on liquid crystal. This is the approach that seems most promising for near-term applications (www.lumotive.com; US Patent 10,451,800; US Patent 10,665,953).*

Our response: We concur with the reviewer that liquid crystal holds advantages in various free-space wavefront control applications, particularly considering the maturity of its relevant chemical and nanofabrication industries. We acknowledge that this technology should have been discussed in our introductory section, and we have now included a comparative analysis. As we pointed out in the following edited paragraph, While appreciating the established success of liquid crystals in various applications, we

recognize that their limitation in tuning speeds may render them unsuitable for certain specific use-cases. This particular constraint has motivated our investigation into devices that integrate other similar but faster organic materials. The modified text is shown below:

Large efforts have been channeled into various modulation mechanisms. Microelectromechanical and nanoelectromechanical systems (MEMS and NEMS) have the advantages of low-cost and CMOS-compatibility, but the speed is limited up to MHz. Phase-change materials have fast, drastic, and non-volatile refractive index change, but lack continuous refractive index tuning and have a limited number of cycles constraining applicability to reconfigurable devices. Through molecule reorientation, liquid crystal can have index modulation over 10%, while under relatively low applied voltages [6,7]. Techniques of LC integration have also advanced after decades of development [8]. However, the tuning speeds are limited to kHz range [6]. Thermo-optic effects can induce relatively large refractive index changes, but the speed is inherently limited and the on-chip thermal management can be challenging.

5. *How to ensure the complete filling of the organic material inside the slot? From previous personal experience, this is not trivial. Bubbles may form in the gap, which may significantly deteriorate the device's performance. This may be verified by taking a cross-sectional image of the device using FIB.*

Our response: The complete filling depends on both the wettability of the organic solution and dielectric device surface. For the solution of the organic electro-optic material, we used 1,1,2-trichloroethane (TCE) as the solvent, which is commonly used in dye coating. The solution concentration was then kept under 9wt% to ensure preservation of the low viscosity and high wettability of TCE. For the Si/SiO₂ surface, we performed solvent and oxygen plasma cleaning prior to the coating. The filled slots showed consistent contrast under both optical microscope and IR camera.

Unfortunately, we cannot perform the FIB on the coated device due to the instrument accessibility. However, many EO-polymer-coated slots in literatures had comparable aspect ratios, and there was no reported incomplete filling (220nm:130nm in [9]; 200nm:40nm in [10]). Thus, we believe that the organics have mostly filled our device and that filling-related problems are not a limitation for the device performance. Finally, we would like to point out that there are some advanced techniques to enhance the infiltration [11]. In the future, these techniques could be used to improve our device's performance too.

[1] Xu, Huajun, et al. "Ultra-high electro-optic coefficients, high index of refraction, and long-term stability from Diels–Alder cross-linkable binary molecular glasses." *Chemistry of Materials* 32.4 (2020): 1408-1421.

[2] Koeber, Sebastian, et al. "Femtojoule electro-optic modulation using a silicon–organic hybrid device." *Light: Science & Applications* 4.2 (2015): e255-e255.

- [3] Abumarshoud, Hanaa, et al. "LiFi through reconfigurable intelligent surfaces: A new frontier for 6G?." *IEEE Vehicular Technology Magazine* 17.1 (2021): 37-46.
- [4] Wang, Haibo, et al. "Performance analysis of multi-branch reconfigurable intelligent surfaces-assisted optical wireless communication system in environment with obstacles." *IEEE Transactions on Vehicular Technology* 70.10 (2021): 9986-10001.
- [5]
- [6] Akselrod, Gleb M., Erik Edward Josberger, and Mark C. Weidman. "Fabrication of metallic optical metasurfaces." U.S. Patent No. 10,968,522. 6 Apr. 2021.
- [7] Chang, Xin, et al. "Electrically tuned active metasurface towards metasurface-integrated liquid crystal on silicon (meta-LCoS) devices." *Optics Express* 31.4 (2023): 5378-5387.
- [8] Gu, Tian, et al. "Reconfigurable metasurfaces towards commercial success." *Nature Photonics* 17.1 (2023): 48-58.
- [9] Kieninger, Clemens, et al. "Silicon-organic hybrid (SOH) Mach-Zehnder modulators for 100 GBd PAM4 signaling with sub-1 dB phase-shifter loss." *Optics express* 28.17 (2020): 24693-24707.
- [10] Heni, Wolfgang, et al. "Nonlinearities of organic electro-optic materials in nanoscale slots and implications for the optimum modulator design." *Optics Express* 25.3 (2017): 2627-2653.
- [11] Taghavi, Iman, et al. "Enhanced poling and infiltration for highly efficient electro-optic polymer-based Mach-Zehnder modulators." *Optics Express* 30.15 (2022): 27841-27857.

REVIEWERS' COMMENTS

Reviewer #1 (Remarks to the Author):

The authors have made a great effort to answer all the reviewers questions and requests. I am fine with their comments and recommend this paper for publication.

Reviewer #2 (Remarks to the Author):

I have read the revised manuscript, as well as the reply to the referees. The authors response to my comments (and those of the other referee) are satisfactory and I recommend publication.